# Evaluating Outdoor Nature-Based Early Learning and Childcare Provision for Children Aged 3 Years: Protocol of a Feasibility and Pilot Quasi-Experimental Design

**DOI:** 10.3390/ijerph19127461

**Published:** 2022-06-17

**Authors:** Oliver Traynor, Paul McCrorie, Nai Rui Chng, Anne Martin

**Affiliations:** MRC/CSO Social and Public Health Sciences Unit, University of Glasgow, 99 Berkeley Street, Glasgow G3 7HR, UK; paul.mccrorie@glasgow.ac.uk (P.M.); nairui.chng@glasgow.ac.uk (N.R.C.); anne.martin@glasgow.ac.uk (A.M.)

**Keywords:** outdoor, nature, early years, childcare, feasibility, play

## Abstract

Systematic reviews have demonstrated the scarcity of well-designed evaluations investigating outdoor nature-based play and learning provision for children in the early learning and childcare (ELC) sector among global Western countries. This study will examine the feasibility and acceptability of the programme and the evaluation design of outdoor nature-based play and learning provision across urban ELC settings in a Scottish metropolitan city. Six ELC settings with different outdoor nature-based play delivery models will be recruited. One trial design will be tested: a quasi-experimental comparison of children attending three different models of outdoor play and learning provision. Measures will be assessed at baseline and five weeks later. Key feasibility questions include: recruitment and retention of ELC settings and children; suitability of statistical matching based on propensity score; completeness of outcome measures. Process evaluation will assess the acceptability of trial design methods and provision of outdoor nature-based play among ELC educators. These questions will be assessed against pre-defined progression criteria. This feasibility study will inform a powered effectiveness evaluation and support policy making and service delivery in the Scottish ELC sector.

## 1. Introduction

The early learning and childcare (ELC) environment is an important setting to support healthy child development. Providing outdoor nature-based play and learning in the early years (0 to 5 years) setting can encourage children to be physically active and develop their social and cognitive skills, while helping them maintain a healthy weight later in life and support more equitable access to local green spaces [1]. However, there is a growing concern that the increasing availability of tablets, smart phones, and television watching, along with a high number of risk averse parents, is culminating in children who are less physically active, who are spending a lot of time indoors, and who have interrupted sleep patterns [2,3,4]. By providing children with more time outdoors in rich green spaces, nature-based ELC could be a key approach for supporting children’s physical, social, and emotional development.

The UK Chief Medical Officers recommend that preschool-aged children (3–4 years) should spend at least 3 h per day physically active, including 1 h in moderate-to-vigorous physical activity (MVPA) [5]. However, research suggests that preschool-aged children struggle to meet the recommendations for MVPA and spend most of their time in low physical activity (PA) [6,7]. MVPA is associated with positive mental health, cardiorespiratory fitness, and body mass index (BMI) in adolescence, therefore it is important to promote these healthy behaviours in the early years [8,9]. Being outdoors is a positive correlate for PA in children [10]. Furthermore, PA, especially outdoor play, is positively associated with most sleep outcomes in preschool-aged children [11]. The natural outdoor environment has a variety of affordances that have the potential to support children’s development of their motor competence, social skills, mental wellbeing, and physical health [10,12,13,14]. A recent review of the literature found positive associations for sedentary time and balance (a key component of motor competence) among children attending nature-based ELC [15]. However, the authors concluded that more high-quality research is required that better describes the nature exposure and outcome measurement tools [15].

Providing children with opportunities to spend time outdoors engaging in unstructured play, especially in nature, is a potentially effective approach for supporting emotional and social resilience and cognitive and physical development [16,17,18]. Through play outdoors, children can engage in a variety of play behaviours that help them learn how to navigate their socio–cultural environments in safe simulations [19]. Furthermore, research suggests that long exposure to natural outdoor environmental lighting (i.e., sunlight) may have a beneficial impact on children’s behavioural and cognitive development [20,21]. The affordances of the natural outdoor environment such as open fields, trees, vegetation, and hilly terrain encourage children to be physically active, imaginative, expressive, and to explore their surroundings, having a beneficial impact on their social skills, mental wellbeing, and cognitive outcomes [1,13,19,22,23]. 

Risk-taking during play exposes children to excitement, uncertainty, and sometimes the possibility of injury [24,25]. Through observations and interviews, researchers have identified eight categories of risky play: play with great heights; play with high speed; play with dangerous tools; play near dangerous elements; rough-and-tumble play; play where children go exploring alone; play with impact; and vicarious play [24,25,26]. By challenging themselves during play, children can manage uncertainties while avoiding excessive risk taking, helping them to develop the skills required to support their increasing autonomy and independence [27]. Nonetheless, although risky play is supported within the Scottish Government’s resources (*My World Outdoors* [28]), support of this type of play behaviour, and others, varies based on practitioners’ beliefs and understanding of outdoor nature-based play and learning provision [29]. The authors are not aware of any attempt in the literature to explore play behaviours and risk-taking opportunities among children attending urban ELC settings in Scotland. 

The role of the practitioner is crucial for supporting quality outdoor play and learning experiences for children, since their beliefs and attitudes influence their practice and values [3,29]. This has implications for how children utilise their outdoor space and the type of outdoor space children have access to (e.g., forest, park, structured outdoor playground). Furthermore, training for practitioners in the provision of outdoor play and learning is not universal, and attitudes and beliefs towards how outdoor nature-based provision should be incorporated into everyday practice varies [30]. Therefore, it is important to understand how practitioners within these ELC settings perceive the ability to incorporate outdoor play and learning within their regular everyday practice.

Most of the research investigating the relationship between nature-based play provision and children’s health outcomes targets older children (greater than 7 years); there is less available evidence for children in the early years (0 to 7 years) within the preschool/kindergarten setting. Much of the available evidence for this population and context suffer from poor methodological quality due to poorly designed evaluations [10,17]. These evaluations are often uncontrolled interventions or cross-sectional studies, with small sample sizes, limited reporting of confounding variables, and poor reporting of the reasons for participant withdrawal, leading to results with a high risk of bias. Evaluations in this field require more sound methodology that can be more effectively achieved by firstly assessing the feasibility of potential evaluation designs.

Feasibility studies have been carried out in British ELC settings before; however, they have focused on interventions specifically targeting obesity prevention through PA and/or healthy eating [31,32,33]. As far as the authors are aware, there has not yet been any formal robust evaluation of outdoor nature-based play and learning provision in the British early years sector. However, such an evaluation has several unique uncertainties that firstly require examination within a feasibility study, including recruiting ELC settings that provide nature-based play and learning, recruitment of participants in these settings, participant randomisation processes, and outcome measures.

In Scotland, all 3- and 4-year-old children (and eligible 2-year-olds) are entitled to 1140 h of funded ELC per year [34]. This is part of the Scottish Government’s commitment to reduce the educational attainment gap and social inequalities in Scotland by providing families with equal access to early years childcare. In 2021, 97% of eligible children were registered for funded ELC [35]. Therefore, the centre-based childcare setting is an opportunity to support the development of lifelong healthy behaviours at the population level.

The ELC sector in Scotland provides several formal settings where play and learning take place. They provide different opportunities for exposure to the outdoors with a variety of activities children can engage in while outdoors. These include: (i).*Fully outdoor* setting, where children spend most of their time in a forest or park with many natural affordances;(ii).*Indoor/outdoor*, where children can move freely between the indoor and outdoor area of their ELC setting; (iii).(*Satellite*, where the ELC has a nature space (e.g., forest or park), but it is not on their physical premises; (iv).*Traditional* ELC setting, often attached to a primary school, where children spend most of their time indoors but with the opportunity to experience the outdoor environment (built and/or natural) as part of structured sessions or play breaks.

The objective of the current feasibility study is to trial a quasi-experimental non-equivocal control design using three models of outdoor play and learning provision in the Scottish early years sector: fully outdoors, satellite, and traditional ELC settings. The aim of this study is to determine whether prespecified criteria associated with the feasibility and acceptability of the program and trial design methods are met sufficiently to progress to a full powered effectiveness evaluation.

## 2. Materials and Methods

The reporting of this protocol has endeavoured to follow the Standard Protocol Items: Recommendations for Interventional Trials (SPIRIT) statement (see Appendix A) [36]. We took a systematic pre-evaluation approach to designing this feasibility study by carrying out two initial phases to identify the most appropriate research questions.

### 2.1. Initial Phases

#### 2.1.1. Phase One: Development of Logic Model

Using secondary data analysis and triangulation methodology, we developed a Theory of Change (ToC) of how nature-based ELC functions in a Scottish urban city, Glasgow. We triangulated interview and focus group transcripts of parents whose children attended nature-based ELC settings in the West of Scotland, observation schedules of children playing outdoors at nature-based ELC settings in the West of Scotland, and international studies investigating the relationship between nature-based ELC settings and children’s health and wellbeing outcomes from several high-income countries. This approach allowed us to explicitly state the inputs, activities, outputs, outcomes, and underlying assumptions associated with the delivery of the programme. Further details of this process can be found elsewhere (Traynor et al., 2022). This process identified key components regarding provision of nature-based ELC that required further investigation with stakeholders before an evaluation design could be developed.

#### 2.1.2. Phase Two: Evaluability Assessment with Key Stakeholders, Refinement of Logic Model, ToC, and Identification of Evaluation Goals

This phase followed a systematic approach to conducting an Evaluability Assessment (EA). The purpose of these EA workshops was to refine the ToC developed in Phase One, ensuring that it was still relevant to the current context (e.g., COVID-19 pandemic, rollout of 1140 h free childcare entitlement), and to identify the evaluation questions that key stakeholders would like answered regarding the provision of nature-based ELC in Scotland. These workshops took place online using MS Teams due to the government COVID-19 guidelines restricting in-person contact at the time.

During analysis of the workshop outputs, several uncertainties associated with the design of an impact evaluation were identified. These included recruitment methods of ELC settings and participants, retention rates of participants, possibility of randomisation, and type of outcome measures. These uncertainties were presented to the research steering group and a decision was made regarding how to address them. The steering group was composed of representatives from the local authority and a third sector organisation involved in the delivery of early learning and childcare in Scotland. The steering group guided the design of the research project, ensuring that the design was appropriate for the early years setting. It was decided that these uncertainties should be addressed in a feasibility study before investing finite resources into a powered effectiveness evaluation.

Six research questions were developed to be addressed in this feasibility study of nature-based ELC. Table 1 outlines these research questions along with how they will be addressed.

### 2.2. Study Design

#### Feasibility and Pilot Study

The development of this feasibility study is based on guidance from the UK Medical Research Council for developing and evaluating complex interventions [37]. This study will pilot a quasi-experimental study design method:

A quasi-experimental non-equivalent control design using propensity score matching with multiple treatment groups. Children who attend two traditional ELC settings will be assigned to the control group and matched with children from two treatment groups: (1) 2 fully outdoors ELC settings and (2) 2 satellite ELC settings. The researchers will follow guidance on implementing propensity score methods with multiple treatment groups (e.g., 2 exposure groups and a control group) [38]. From baseline to follow-up, the study duration will be 5 weeks. Play and learning will continue as normal in both settings and the outcomes mentioned below will be measured to assess the possible impact of outdoor play and learning.

This study will employ a mixed-methods approach. Both quantitative and qualitative data will be collected to address the research questions. Alongside the piloting of these two study designs, we will carry out a process evaluation using semi-structured interviews. Qualitative methods are considered an important methodological approach to understanding the causal mechanisms of complex health interventions [39].

### 2.3. Ethics Approval

Ethical approval has been provided by the College of Social Science, University of Glasgow (application number: 400210145) and Glasgow City Council (reference number: 21.26).

### 2.4. Setting

Glasgow is the largest metropolitan area in Scotland and one of the most socioeconomically disadvantaged areas in Western Europe, with more than a third of the cities’ children estimated to be living below the poverty line [40]. The Glasgow City Council (GCC) area has one of the highest numbers of nature-based ELC settings in Scotland, with at least 18 ELC settings registered or in the process of becoming registered by the Care Inspectorate as a provider of a specific nature-based ELC model [41].

The GCC area has 110 ELC settings that are operated by the GCC Education Services and over 300 ELC settings that are privately or voluntary operated. Within these settings, there are several that are registered with the Care Inspectorate as a satellite model or are in the process of becoming registered. There are around six functioning as fully outdoor nature-based models in the GCC area, with more across Scotland. Many of these are private and voluntary childcare providers, with some functioning in partnership with GCC. Partnership providers are social enterprises that work collaboratively with GCC to extent their ELC provision. The Scottish Government’s 1140 h of free childcare entitlement is applicable to all ELC providers with the caveat that some private and partnership settings request that parents cover some additional costs. Furthermore, each setting is based in a different area of Glasgow, uses different types of green/natural spaces, and varies in size with regards to the number of attending children and practitioners. All of these factors, in addition to context, dose, and practice will influence the daily operations of each ELC setting and thus will be investigated in this feasibility and pilot study.

In the GCC local authority ELC settings, there are between 20 and 100 three- to five-year-olds registered at any one time with an average of 60 or 38 children per setting, depending on whether the ELC setting operates 50 or 38 weeks per year, respectively. In the private, voluntary, and independent sector, ELC settings have between 12 and 80 three- to five-year-olds registered, with an average of around 30 children enrolled at any one time.

### 2.5. Participants

ELC settings in the GCC area, representing the different models previously mentioned, will be invited to participate in the study, including their headteacher/manager and 2 practitioners. Figure 1 illustrates the study flow diagram.

#### Inclusion and Exclusion Criteria

The inclusion and exclusion criteria (Table 2) are to ensure that children who are enrolled in the study are as similar as possible to children who would be enrolling in the ELC setting for the first time at the beginning of the academic year (the timepoint at which a full-scale evaluation would be recruiting). Eligible children are those who attend these ELC settings, are 3 years old, and have consent from their parent/carer to participate. Participants will receive a voucher of GBP £10 as a token of appreciation for their participation. Participating ELC settings will receive a GBP £100 donation and an additional GBP 10 for each ELC educator that participates in the interviews.

### 2.6. Sample Size

As this is a feasibility/pilot study, no sample size to detect between group differences was calculated. The study may provide preliminary data for the calculation of a sample size for a future full-scale evaluation. We aim to recruit 2 fully outdoor ELC settings, 2 satellite ELC settings, and 2 traditional ELC settings.

A minimum of 10 children per ELC setting will be recruited. Two practitioners and one headteacher/manager will be recruited from each ELC setting.

### 2.7. Recruitment

All eligible ELC settings will be contacted via email using publicly available email addresses and phone numbers. Emails will contain documents with details on the study, including the methods to be used and what will be asked of ELC settings that participate. Those that express an interest in taking part through completing an Expression of Interest Form will be followed up with a participant information sheet (PIS) and consent form. The PIS will provide further details of the methods to be used in the study and how they affect individual participation. If headteachers/managers and practitioners wish to participate, they can return the completed consent form to the research team via email. Participating ELC settings will be given study flyers to distribute to the families that attend their ELC setting. The flyers will contain a QR code for parents/carers to scan for further information on the study. The QR code will take parents/carers to a secure University of Glasgow website that has a participant information sheet and consent form for parents/carers to complete. If parents/carers would like their child to take part in the study, they can complete the consent form and email it back to the research team.

Propensity Score Matching

The propensity score is the probability of being exposed given the values of measured confounding variables [42]. In the same manner that randomisation will on average lead to measured and unmeasured covariates being balanced between study groups, assigning participants based on their propensity score will on average lead to measured baseline covariates being balanced between the study groups [43]. Our propensity score model will include all measured baseline covariates collected in the demographic survey. Participants from the traditional ELC settings (comparison) will be matched with participants from the fully outdoor settings and satellite settings (exposure) with similar propensity scores. We will use nearest neighbour matching within a specified caliper distance that is proportional to the standard deviation of the recorded covariates [43]. Once matched, the treatment effect will be determined by directly comparing the outcome data between matched samples of the exposure and comparison groups. Researchers have shown that the use of propensity score matching with small study samples can demonstrate unbiased estimations of treatment effect if the appropriate confounding variables are included in the propensity score model [44].

### 2.8. Measures

The feasibility questions that this study will address are shown in Table 1. **RQ1** intends to address the uncertainties associated with recruitment and retention of ELC settings and study participants. The following data will be collected across all participating ELC settings:Number of eligible ELC settings that were approached to participate in the study.Number of ELC settings that expressed an interest in taking part, number of ELC settings that declined an invitation to take part, and number of ELC settings that did not respond.Number of eligible children who were approached to participate in the study.Number of children whose parents consent for them to participate and number of children who did not have consent to participate.Number of participating children who leave the ELC setting after the study has begun.Attendance records of participating children will be collected throughout the study timeline to determine retention rates.

**RQ2** addresses whether matching on propensity score is feasible to detect changes in outcomes between exposure and comparison groups. This will be measured by:Number of children assigned a propensity score.Number of successful matches using propensity score.

A powered effectiveness evaluation can only be successful if its study design is able to detect differences in children’s health and wellbeing outcomes as a result of being exposed to outdoor nature-based play and learning. 

Addressing **RQ3** will identify which outcomes and measurement tools should be taken forward to the next stage of evaluation. This will be measured by:The completeness of the measurement assessments from baseline to follow-up.Which measurement tools demonstrate a positive, negative, or null effect between exposure and comparison groups from baseline to follow-up.The acceptability of the measurement tools within the study population.

The outcomes and their measurement tools can be found in Table 1. 

By examining the monitoring and evaluation (M&E) tools that are currently in place across ELC settings, we can determine whether similar outcomes are recorded among all participating ELC settings (**RQ4**); if so, it might be possible to develop a standardised method of analysing changes in outcomes. This approach could then be used in a powered effectiveness evaluation. **RQ4** will be measured by collecting a sample of M&E tools per recruited sample (*n* = 5), recording the outcomes that are monitored, and identifying whether analysis can be standardised.

**RQ 5 and 6** will be assessed through a **process evaluation** using qualitative interviews. Acceptability of the programme and study design methods will be measured based on the number of major barriers discussed during semi-structured interviews with ELC headteachers and practitioners. This will be conducted using the four components of the Normalisation Process Theory (NPT) [45]. See Appendix B for the interview guide and NPT components.

NPT identifies factors that support and impede the normalisation of complex interventions into routine practice [46,47]. The theory centralises on the work that individuals and groups do to allow an intervention or programme to become normalised. NPT has four primary components: *coherence* (sense making, such as how easy is it to understand the purpose of outdoor nature-based play and learning?); *cognitive participation* (or engagement, such as are practitioners committed to providing outdoor nature-based play and learning, are they supportive of the study recruitment methods?); *collective action* (are practitioners sufficiently trained to provide outdoor nature-based play and learning, are they committed to ensuring children remain in their study groups?); and *reflexive monitoring* (formal and informal appraisal of the programme and evaluation design). By using NPT, we can optimise the study design methods (recruitment, outcome measures) by determining whether the pilot study design was acceptable for practitioners and headteachers/managers. NPT will also help us understand to what extent outdoor nature-based play and learning is considered a part of everyday practice at participating ELC settings.

The progression criteria for RQs are discussed in the next section.

#### 2.8.1. Progression Criteria 

The CONSORT 2010 extension to randomised pilot and feasibility trial guidelines recommend setting out progression criteria when reporting feasibility and pilot studies [48]. In line with recommendations, our progression criteria will be assessed using a traffic light system with varying levels of acceptability, rather than strict thresholds [49,50,51]. For example, GREEN, strong indication that study design and programme is feasible and can be taken to the next stage of evaluation; AMBER, study design and programme could be feasible with some modifications; and RED, study design and programme should not progress forward without serious consideration and modification. The progression criteria for each RQ can be found in Table 3.

As demonstrated in Table 3, the progression criteria for **RQ1** have been informed by previous feasibility studies in this setting, where researchers have demonstrated a recruitment rate of 32%, 37%, and 10% of eligible ELC settings, respectively [31,32,33]. Our recruitment rate of at least 30% of contacted eligible ELC settings will demonstrate that our recruitment methods for ELC settings are feasible for a larger effectiveness evaluation. Moreover, to determine the feasibility of our participant recruitment methods, we aim to have at least 50% of eligible children at fully outdoor settings return a signed consent form and 25% of eligible children at all other settings return a signed consent. These estimates are different because we believe buy-in from parents at fully outdoor ELC settings will be greater than some of the traditional and satellite settings given the nature of the research project (exploring the impact of children playing outside). Additionally, parents choose to enroll their children at fully outdoor ELCs, whereas places at council-organised settings are assigned. Finally, if 80% or more participating children are retained at the study follow-up time period, the participant recruitment methods will be considered feasible to progress to an effectiveness evaluation. If less than 50%, the pilot study will be halted and will return to the design stage. If retention rates are between 50% and 79%, then the retention methods will be reviewed to determine whether they can improve before progressing to a powered effectiveness evaluation. 

The progression criteria for **RQ2** are also conditional upon which traffic light criteria are achieved regarding retention rates within **RQ1**. The number of retained children will undergo matching based on their propensity score. 

The traffic light criteria for **RQ3** are based on what we determine to be sufficiently acceptable to carry out analysis of the outcome data to detect an effect if one does exist. Outcome measures that can detect a signal in effect will be considered for progression to the next stage of evaluation subject to meeting the other progression criteria. These will also be considered alongside the number of barriers mentioned related to the measures in the interviews and feedback from parents/carers. The outcomes and their measurement tools are outlined in Table 4.

The traffic light criteria for **RQ4** are based on the extent to which analysis of M&E tools can be standardised across ELC settings. It is important for researchers to maximise the use of M&E tools that are already in place to reduce the burden on practitioners, increasing the likelihood of optimal participation and supporting evaluation capacity.

Finally, the progression criteria for **RQs 5 and 6** are based on the qualitative findings informed by NPT. If participants feel they have agency and resources to support outdoor play and learning and there are no major barriers associated with the study design, then progression to a powered effective evaluation will be possible after consultation with the research steering group.

#### 2.8.2. Measurement Tools

Table 4 demonstrate the measurement tools that will be used at each timepoint. Although one timepoint is sufficient for determining the feasibility of collecting the data by researchers, there are other uncertainties that can be addressed by using two timepoints. Collecting data at both timepoints will allow us to assess the acceptability of the study design by estimating the burden of taking part in the measurements for children and also the ELC educators supporting the data collection. Additionally, we will be able to estimate potential programme effects and determine which child health outcomes should be taken forward as primary and secondary outcomes in a powered effectiveness evaluation.

#### Demographic Questionnaire

At baseline, parents/carers of participating children will be asked to complete an online demographic questionnaire to describe their family background (child’s date of birth, gender identity, ethnic identity, number of siblings, double or single parent household, age of mother when child was born), as well as how many hours per week the child spends at their ELC setting and their home postcode. The postcode will be used to calculate the level of multiple deprivation experienced at the local area level where the child lives as defined by the Scottish Index of Multiple Deprivation (SIMD) [52]. The SIMD is a composite measure of education, crime, health, income, and housing to develop an estimate of area-based deprivation for all neigbourhoods in Scotland. The SIMD scores for participating ELC settings will also be calculated. 

#### Strength and Difficulties Questionnaire (SDQ)

Emotional and behavioural wellbeing will be measured using the validated parent-reported SDQ [53,54]. The parent-reported SDQ was chosen rather than the teacher-reported SDQ to reduce the burden on participating ELC settings. The SDQ has 25 items divided into 5 scales with 5 items each: emotional symptoms, conduct problems, hyperactivity/inattention, peer relationship problems, and prosocial behaviour. Parent/carers will be asked how true different statements are about their child on a 3-point scale ranging from 0 (not true) to 2 (certainly true). The total difficulties score is calculated based on all the scales excluding the prosocial scale to give a score between 0 and 40. An online version of the SDQ will be completed before the study exposure period commences and at the end of the study exposure period (follow-up).

#### Height and Weight

Children will have their height and weight measured at baseline across all participating ELC settings (to the nearest 0.1 cm/kg). Weight status will be presented as a BMI z-score. Children in the fully outdoor and traditional ELC setting will also have their height and weight measured at follow-up.

#### Preschooler Gross Motor Quality Scale (PGMQS)

Some research suggests that exposure to play and learning in nature-based ELC may have a greater benefit on certain motor skills than play and learning at traditional ELC settings alone. These associations could be more applicable to gross motor and vestibular skills, rather than object control [10]. Development of these motor skills provide the foundation for more specialised movements that can influence long-term PA levels. In our stakeholder workshops, the ELC practitioners identified several activities that are popular among children attending outdoor nature-based settings such as climbing and managing obstacles such as trees or those built by children themselves. These require important motor skills such as the ability to balance. This study will apply the balance assessment of PGMQS, a validated tool for use in 3-year-olds that assesses four components of balance: single leg standing, tandem standing, walking line forward, and walking line backward [52]. This tool has previously been used to investigate balance development in preschoolers taking part in an outdoor loose parts intervention in Nova Scotia [55]. One child will be assessed at a time; the researcher will demonstrate how to correctly perform the task and then ask the child to perform the task. They will have one practice trial followed by two scored trials. Each balance task is composed of 4 or 5 criteria as outlined in Table 5. Each criterion will be scored a 0, indicating the movement performed incorrectly, or 1, indicating the movement performed correctly, to get a total balance score out of 18. The two scored trials will be added to give a total score out of 36. Measures will be collected at baseline and follow-up (5 weeks).

#### Physical Activity, Sedentary Time, and Sleep

Children’s physical activity, sedentary time, and sleep will be measured using a wrist-worn triaxial accelerometer (Axivity AX3). Wrist-worn devices have been found to be more acceptable and less burdensome compared with hip-worn devices [56]. Preliminary piloting of the Axivity devices with a small sample of 3-year-olds for 3 days has found high levels of acceptability and compliance. Children will be asked to wear the activity monitor on their non-dominant wrist to limit miscalculated activity counts during sedentary behaviours (e.g., drawing, writing, and playing on mobile electronic devices) [57]. The devices will be configured to record raw acceleration data using Open Movement GUI (OMGUI, V1.0.0.43). Parents/carers will be asked to ensure that their child wears the device for 7 days (24 h/day) with a minimum of 3 consecutive days, removing it only if it causes discomfort or when the child is in water (e.g., swimming or bathing). Parents/carers will be provided with an activity diary to log any time the device is removed from their child’s wrist, the reason why, and the time it is placed back on their child’s wrist. The devices will be programmed to start at the end of the day when anthropometric measures and motor skills assessment are completed. Age-appropriate cut points will be applied to the data to identify activity intensities. Physical activity intensity cut points vary depending on the type of activity monitor used (e.g., ActiGraph, GENEActive, Axivity), the location on the body where it is mounted (e.g., wrist, waist, or thigh), and the age group of the study population. There is not yet a consensus regarding a standard approach to analysing PA by accelerometer [58,59]. Moreover, Axivity monitors have not yet been validated in the preschool-aged population [60]. Research suggests that due to the sporadic activity pattern of preschool-aged children, shorter epoch lengths (5-s epoch) at a sampling frequency of 100 Hz may be more suitable to identify very short periods of movement [61]. Therefore, based on recent research using wrist-worn accelerometers with preschoolers, our PA intensity thresholds are: sedentary behaviour, ≤221 counts per 5-s epoch; light PA, 222–729 counts per 5-s epoch; MVPA ≥ 730 counts per 5-s epoch; and total PA ≥ 222 [62]. These cut points have been used in wrist-worn accelerometers in a similar Scottish ELC setting [63]. The light sensor on the Axivity monitor is a logarithmic lux sensor (a measure of the intensity of light) that has a wavelength characteristic similar to the human eye. It is recommended that the AX3 monitor is calibrated at 1000 lux [64].

Sleep will be estimated by calculating the sleep period time (SPT) frame, the time window from initial sleep onset and waking up after the last sleep episode of the night, based on z-angle variance [65,66]. Using this, the average time of sleep onset and waking (beginning and end of SPT window) will be calculated. The total period of continuous inactivity periods (no change in z-angle of >5° for a minimum of 5 min) within the SPT-window will be calculated to estimate sleep duration per night, then averaged across available nights [66]. A minimum wear time of 3 complete days will be required for analysis. There is not yet a standard recommendation for non-wear time among preschool-aged children [61]. Therefore, we will define the device as not being worn if there are 60 consecutive minutes of zero acceleration recorded. This will be cross-checked with the activity logs to identify any non-wear time less than 60 min.

#### Play Behaviour

A base map of the outdoor space (a geographical mapped representation) used by each ELC setting will be developed for recording children’s play behaviours using Google Maps. A base map allows observers to document the physical features and layout of the outdoor environment such as vegetation, play structures, and pathways [23]. The Tool for Observing Play Outdoors (TOPO) will be used to code pre-determined play types onto the base map of the outdoor space [19]. This study will use a place-based protocol to assess the quality of the environment for supporting play types [23]. A pre-defined observation zone will be scanned clockwise until a play event is detected. The researcher then observes the child(ren) for around 15 s, records the play behaviours, and maps the event onto the base map. Afterwards, scanning of the observation zone begins again from the point that it stopped, until another play event is detected. Only data from children who have consent from their parent/carer to participate will be recorded. The expanded TOPO-32 version will be used to assess the feasibility of fully completing the data collection tool. The tool has 9 primary play types: physical play, exploratory play, imaginative play, play with rules, bio play (interactions with the natural environment), expressive play, restorative play, digital play, and non play, along with 32 associated subtypes. The TOPO-32 assigns primary-subtype combinations for each observed play episode alongside peer, adult, and environmental interaction codes. For each play episode, two primary-subtype combinations will be recorded. Additional categories will be added to the observation tool such as risk taking during play. Our observation protocol will use the play categories defined by Sandseter and colleagues (Table 6) as these are considered the most suitable approach for identifying the risk-taking affordances [24,25,26]. Appendix C has an example of the observation protocol to be used in the study.

#### Semi-Structured Interviews

Using a semi-structured interview guide (Appendix B), developed using the Normalisation Process Theory, acceptability of the programme and study design methods will be assessed with ELC headteachers and practitioners. Interviews will cover participants’ views on the implementation of nature-based ELC (e.g., staff support, access to training and resources, contextual factors that influence implementation such as weather) and acceptability of the study design methods (how easy/difficult ensuring children remain in assigned groups, are data collection methods too intrusive, did the presence of an external researcher impact delivery of the programme in any way). Interviews will last one hour and take place in person or over the phone, whichever is most convenient to the participant. Interviews will be audio recorded and transcribed verbatim.

#### Monitoring and Evaluation Tools

Our EA workshops identified several data collection methods that are already in place within ELC settings to record children’s progress throughout their time at the settings. These are known as online journals. A sample of these data collection methods will be requested from participating children. The online journals will be examined to check for similarities of the type of information recorded and to determine whether standardised analysis may be implemented across the online journals from each ELC setting. A sub-sample of 5 participants from each participating ELC setting will be selected at random. OT will ask the headteacher/manager for access to the information collected within the online journals of children who have consent from their parents. An Excel spreadsheet will be developed based on the information extracted from the online journals. This information will include what outcomes are recorded, how many times, and by whom.

## 2.9. Analysis

### 2.9.1. Data Management

All data will be stored in a secure storage system at the Social and Public Health Sciences Unit (SPHSU), University of Glasgow. The raw data will only be accessible to the immediate research team. Consent forms will be stored separately from participant data and a unique identification code will be assigned to each participant and ELC setting. Raw data such as interview recordings and identifiable transcripts will be securely deleted at the end of the project timeline. The de-identified data will be stored for up to 10 years and be available to researchers who are interested and have relevant ethical approval in an anonymised format in line with the University of Glasgow retention policy and general data protection regulation.

### 2.9.2. Recruitment

The feasibility of the recruitment methods and retention rates will be determined by calculating the proportion of children measured at baseline and follow-up. Participant characteristics of those who complete the study and those who drop out will be investigated to determine any sources of bias. 

### 2.9.3. Statistical Analysis

Baseline characteristics of the children and ELC settings will be summarised descriptively using means and standard deviations. For outcome measures: medians and interquartile ranges will be calculated for continuous variables and frequencies and percentages for categorical variables. The summary statistics will be examined alongside demographic variables to inform the sample size and recruitment methods for the full-scale evaluation. Comparisons will be made by the SIMD level of ELC settings. Any missing data will be described to inform the full-scale evaluation. STATA statistical software will be used for all analyses.

Exploratory inferential analysis using ANOVA and linear regression will be performed within the study population of each ELC setting. Analysis will include a covariate adjustment based on the confounding variables collected in the demographic survey (e.g., gender, SIMD score, ethnicity, etc.) and covariate adjustment using children’s physical activity levels, since higher levels of PA are known to positively influence mental wellbeing.

Using propensity score matching, we will estimate the average treatment effect (ATE) of attendance at nature-based ELC on each health and wellbeing outcome (e.g., PA). A logistic model will predict each participant’s propensity score using the covariates collected in the demographic survey. The ATE on an outcome will be estimated by matching participants to another participant whose propensity score is within the pre-specified caliper distance.

### 2.9.4. Qualitative Analysis

The interviews will be audio recorded and transcribed verbatim by an approved transcription service. Thematic analysis will be applied to code the data based on the pre-determined process evaluation topics (feasibility components) [67]. NVivo 12 will be used to analyse the transcripts (QSR International Pty Ltd., 2020). Transcripts will first be read and initial codes created based on the interview questions and research aims. OT will develop a preliminary coding framework and discuss this with PM, NRC, and AM. Themes and sub-themes will be created both deductively and inductively, while making sure the theme labels remain representative of the data. To reduce bias, a sub-sample of the transcripts will be reviewed by a co-author and any discrepancies will be discussed and resolved. After all themes have been created, matrices will be developed to view responses and frequency of themes amongst the transcripts. To ensure rigour in the research, OT will engage in reflexive thinking throughout the research process. This will include discussing uncertainties with co-authors and constantly returning to the literature to elucidate particular themes or experiences. OT will endeavour to have spacing between interviews to allow for reflection and learning from each interview. To encourage participants to feel comfortable, interviews will be arranged for mutually convenient times and (if applicable) locations.

## 3. Discussion

The evidence base underpinning the effectiveness of nature-based play and learning provision suffers from poor methodological quality caused by a lack of well-designed evaluations [10]. We argue that evaluations can be better designed if a feasibility and pilot study is first carried out to address key uncertainties that may compromise a fully powered effectiveness evaluation. Feasibility studies are important for estimating recruitment and retention rates, data collection procedures and analysis, and acceptability of the programme [50]. The recent updated Medical Research Council (MRC) guidance on evaluating complex interventions recommends that key uncertainties should be investigated in a feasibility study and assessed against predefined progression criteria associated with the evaluation design and acceptability of the programme [37]. This in turn will ensure that finite resources are not wasted on under-powered large-scale evaluations. As outlined in this paper, this study will address key feasibility questions identified through programme theory work (Phase 1; Traynor et al., 2022) and collaborative engagement with key stakeholders (Phase 2; Evaluability Assessment). These include, recruitment and retention procedures, outcome measurement methods and analysis, and statistical matching procedures. Propensity score matching is a valuable method when randomisation is not possible within the study contexts [38]. Propensity score matching reduces the impact of bias on the outcomes under investigation by controlling for measured covariates. Nonetheless, this method does have its limitations such as the possibility of unmeasured covariates influencing the outcome of interest. Furthermore, our embedded process evaluation demonstrates a reliable theory-based approach for determining the acceptability of the study design methods and provision of the programme using NPT. Moreover, having pre-specified progression criteria based on a traffic light system for each research question demonstrates an explicit process to decide whether to proceed, proceed with modifications, or not to proceed for each trial procedure and programme [68].

Beyond addressing key feasibility aspects, our findings will have implications for the wider field of nature-based ELC research and any future fully powered effectiveness evaluation of outdoor nature-based ELC in Scotland. At present, evaluations often have a poor description of the dose and quality of their nature exposure element, making it difficult to determine the pathways by which nature-based ELC influence child health outcomes [15,18]. The present study has three clearly defined models of ELC (traditional, fully outdoors, and satellite) that differ in their provision of outdoor play and learning (e.g., time spent outdoors, number of children outdoors per session, and outdoor space used). Additionally, our observational methods and interviews will provide further detail regarding the affordances of each outdoor location with regards to environmental features and play behaviours. Therefore, alongside Gibson’s Theory of Affordances [69], these procedures will help a full-scale evaluation develop hypothesised pathways with outcomes to identify an effect, if one does exist. Furthermore, the role of the practitioner is also crucial within the provision of nature-based ELC [29]. Recent research has highlighted the need for more systems-based approaches with ELC practitioners to identify the factors important for implementing nature-based ELC (Zucca et al., under review). Our interviews with ELC practitioners and headteachers will contribute to our understanding of the specific pedagogical practices and contextual factors that influence how ELC educators support children’s outdoor nature-based play and learning experiences in nature. Finally, our findings will help policy makers and local authority decision makers optimise the resources they have by encouraging reflection on their current practice.

## 4. Conclusions

To support the implementation of outdoor nature-based ELC provision it is important for researchers to work collaboratively with practitioners and policy makers across the early years sector. This paper has demonstrated how we have developed a feasibility and pilot study to evaluate the provision of outdoor nature-based ELC, informed by collaborative engagement with key stakeholders and a research steering group. Our findings will demonstrate whether a quasi-experimental study design using propensity score matching is feasible and acceptable to take forward to a fully powered effectiveness evaluation. Furthermore, we will be able to identify to what extent outdoor play and learning provision has been normalised within early years practice and whether there are any key barriers to further normalising the provision of outdoor nature-based play and learning within Glasgow’s ELC settings. By involving our steering group within the decision-making process, these findings can help inform the implementation and subsequent evaluation of nature-based ELC settings across other parts of Scotland and the UK. 

## Figures and Tables

**Figure 1 ijerph-19-07461-f001:**
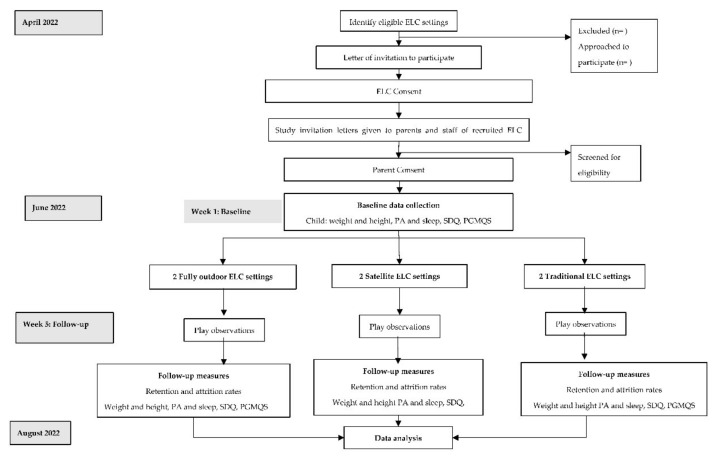
Study flow diagram. ELC—early learning and childcare; PA—physical activity; SDQ—strengths and difficulties questionnaire; PGMQS—preschool gross motor quality scale.

**Table 1 ijerph-19-07461-t001:** Outline of criteria, data collection tools procedures, study population, and analyses required for each research question.

Research Question	Criteria (How We Know We Have Achieved Objective) and Information Required	Data Collection Tools	Procedures	Study Population	Analyses
1. To what extent are the intended participants ***recruited*** to the study and ***retained***?	Data collected on eligible ELC settings, number that consent to take part, do not consent, and do not respond. Demographic characteristics, number that are retained or lost at follow-up	A map of ELC settings in GCC area will be used to identify eligible ELC settings. Email will be used for expression of interest. Setting demographics will be acquired via questionnaire. Parents will receive parent information sheet (PIS), consent form, and demographic characteristics questionnaire.	OT will approach eligible ELC settings via email with PIS, consent form, and letter of agreement, and a request for languages spoken at the nursery. These will be signed and returned if settings wish to take part. Flyers with QR codes will be distributed to parents. QR codes will give parents access to PIS, consent form, and surveys. Paper copies will also be available. The number of children who consent, do not consent, and do not respond will be recorded based on the number of returned consent forms. The number of retained participants and ELC settings will be recorded based on the number still participating at follow-up	Participating ELC settings in Glasgow and the children and staff attending them.	Calculate the number of consenting participants as a percentage of the total eligible ELC settings and children.
2. To what extent can ***propensity score matching*** based on multiple treatment groups be suitably applied within this context?	Data collected on number of children successfully matched across all participating ELC settings.	Children’s demographic information used for matching.	Observed covariates are recorded in both groups. A propensity score is calculated based on these characteristics. Propensity score of children in fully outdoor settings and satellite settings will be matched with propensity score of children in traditional ELC settings. The difference in average outcomes of the matched pairs are compared and a local average treatment effect is estimated. To reduce the risk of bias, matching will be combined with difference-in-differences method, facilitates correction of differences between groups that are fixed over time (reduces the risk of bias in the estimation).	Participating children in all participating ELC settings in Glasgow.	Average treatment effect will be calculated based on the propensity scoring matching of participants from each group.
3. What ***outcomes*** should be included in a future impact evaluation and how should they be measured?	Data collected on outcome measurements using, device worn measurements, questionnaires, and observations. Completeness of data collected and ease of use regarding measurement tools. Which outcome measurement tools identify a change in effect from baseline to follow-up (positive, negative, or null)?	Weight and height: portable standiometer and digital scales. Motor competence: Preschool Gross Motor Quality Scale (PGMQS)PA and sleep: Axivity device. Social and emotional outcomes: strength and Difficulties Questionnaire (SDQ). Play: Tool for Observing Play Outdoors (TOPO). Semi-structured interviews.	Baseline: weight and height, PGMQS, Axivity (7 days with a minimum of 3 full consecutive days), SDQ. TOPO will be implemented during exposure period. Follow-up: weight & height, PGMQS, SDQ, Axivity. Semi-structured interviews with managers and practitioners. Parents will be given a questionnaire at the end of the study asking how the SDQ and Axivity measurement tools were received.	Participating ELC settings in Glasgow, children, staff, and parents/carers.	Summary statistics will be presented for the outcome measures, means and standard deviations will be presented.
4. To what extent are current monitoing and evaluation **(*M&E) tools*** used within, and standardised across, ELC settings?	Monitoring and evaluation tools currently used at participating ELC settings will be examined to determine whether they can support measurement of outcomes and if analysis can be standardised across settings.	Learning journals (see-saw)	A sample of journals from each participating ELC setting (journals of participating children) will be collected and analysed for similar themes (e.g., are the same outcomes recorded, how often is information recorded in the journals, does the same practitioner record for the same child each time).	Participating ELC settings and participating (consent) children.	Thematic analysis to determine with a standardised framework can be developed for analysing journals to support outcome monitoring in a full-scale evaluation.
5. To what extent is the ***programme acceptable*** to ELC managers and practitioners?	Acceptability of programme implementation.	Semi-structured interviews with ELC managers and practitioners.	Purposive sample of managers and practitioners will be invited to interview, consent obtained, interviews recorded and transcribed.	Participating ELC settings, managers, and practitioners.	Qualitative interviews will be thematically analysed.
6. To what extent is the ***study design acceptable*** to ELC managers and practitioners?	Acceptability of trial methods including recruitment process and data collection methods.	Semi-structured interviews with ELC managers and practitioners.	Purposive sample of managers and practitioners will be invited to interview, consent obtained, interviews recorded and transcribed.	Participating ELC settings, managers, and practitioners.	Qualitative interviews will be thematically analysed.

**Table 2 ijerph-19-07461-t002:** Inclusion and exclusion criteria for the feasibility study of nature-based ELC.

Participant	Inclusion Criteria	Exclusion Criteria
ELC settings	Local childcare providers (nurseries, family learning centres, and preschools) in the GCC local authority area. The research team has a list of all ELC settings in the GCC area with their assigned data zone from the Scottish Index of Multiple Deprivation (SIMD, 2020). ELC settings that operate as traditional, satellite, or fully outdoors.	Childcare settings such as creches, child minders, playgroups, and au pairs/nannies. Not located in Glasgow, Scotland.
Children	Three years old at the time of recruitment or turning three during the study period (May to August 2022). Must spend at least three sessions per week (morning or afternoon sessions or all day) at the ELC setting included in the study. Children who have consent from their parents to participate. Specific to satellite settings: children who have attended their satellite outdoor space 3 times or less.	Not three years old at the time of recruitment or will not be three before the end of the study period (August 2022). Spend less than 3 sessions per week at the ELC under study. Have a serious injury or disability (e.g., wheelchair bound, broken leg, restricted arm movement) that would significantly limit their ability to engage in the study measurement methods. Children whose parent/carer does not provide informed consent to take part. Specific to satellite settings: children who have attended their satellite outdoor space more than 3 times.
ELC educators	Practitioners who supervise the children and support their play while outdoors. Managers/headteachers of participating ELC settings.	Practitioners who do not spend their working hours outside with the children when the children are outside (e.g., administrative staff).

**Table 3 ijerph-19-07461-t003:** Study criteria for progression to a powered effectiveness evaluation.

Feasibility and Pilot Study Criteria for Progression to a Powered Effectiveness Evaluation				
Research Question	Traffic light progression criteria	Recommendation if Green, Amber, or Red	Method of assessment	Rationale
1. To what extent are the intended participants ***recruited*** to the study and are they ***retained***?	Recruitment: *ELC Settings* GREEN: at least 30% of contacted ELC settings express a willingness to participate. AMBER: 10 to 29%. RED: less than 10% of contacted ELC settings respond. *Participants* GREEN: fully outdoor ELC settings, at least 50% of eligible children return a signed consent form. At satellite and traditional ELC settings, at least 25% of eligible children return a signed consent form. AMBER: at least 50% of fully outdoor ELC settings achieve their 50% recruitment target. At least 50% of satellite and traditional ELC settings achieve their 25% recruitment target RED: Amber target is not achieved	GREEN: Strong indication to use the same recruitment process in full effectiveness evaluation. AMBER: Indication that recruitment process might work, but should be discussed with research steering group. RED: Indication that recruitment process needs serious revision before full effectiveness evaluation.	Data collected on eligible ELC settings and participants, response rates, and non-response rates.	Based on recruitment and retention rates of past feasibility studies in UK early years settings (Barber et al., 2019; Kipping et al., 2019; Malden et al., 2019).
	Retention GREEN: 80% or more participant retention rate. AMBER: 50–79% participant retention rate. RED: less than 50% participant retention rate.	GREEN: Strong indication to use the same recruitment process in full effectiveness evaluation. AMBER: Indication that recruitment process might work, but should be discussed with research steering group. RED: Indication that recruitment process needs revision before full effectiveness evaluation.		
2. To what extent can propensity score matching based on multiple treatment groups be suitably applied within this context?	GREEN: 80% or more enrolled children suitably matched based on propensity scores. AMBER: 50–79% enrolled children matched. RED: less than 50% enrolled children matched.	GREEN: Strong indication that matching children based on their propensity score is a reliable comparison method to be used in a powered effectiveness evaluation. AMBER: Indication that the matching method might work in an effectiveness evaluation; however, it should be considered alongside other criteria such as the level of missing data. RED: The study design method needs careful consideration before being used again.	Determine the suitability of the covariates collected in the demographic survey are sufficient to calculate reliable propensity scores for matching.	We consider 80% to be achievable if the criteria regarding retention in RQ1 is successful.
3. What ***outcomes*** should be included in a future impact evaluation and how should they be measured?	GREEN: 70% or more measures are returned fully completed at baseline and follow-up. Outcome measures are able to identify a change from baseline to follow-up and a difference in effect between intervention and comparison groups (positive or negative). No major acceptability issues during educator interviews or via parent feedback. AMBER: Outcome measure completion rate of 60% or more. An indication that there may be a difference in effect but insufficient data to determine definitively. Three or four major acceptability issues raised by interviewees or through parent feedback; however, mitigating strategy identified. RED: Less than 60% of outcome measures completed. Major acceptability issues raised regarding measurement methods with no possible mitigation strategy.	GREEN: Strong indication to proceed with the outcome measurement methods. AMBER: Medium indication to proceed. Recommend discussing the measurement methods with research steering group. RED: Indication of doubt as to whether to proceed. Measurement methods should be discussed with steering group, taking into consideration findings from RQ6 and whether different measurement tools might be better suited to detect an effect in outcomes.	Baseline and follow-up outcome measures. Feedback from parents in activity diary. Process evaluation interviews with ELC educators.	We consider 70% completion rate sufficient to carry out analysis.
4. To what extent are current ***M&E tools*** used within, and standardised across, ELC settings?	GREEN: a standardised method of analysing M&E tools across ELC settings is identified and can be used for measuring specific outcomes in a powered effectiveness evaluation. AMBER: there is the potential to standardise the analysis of M&E tools across settings that would require minor changes within ELC practice. RED: there is considerable variation between ELC settings regarding the recording methods of outcomes and adapting practice would be counterproductive.	GREEN: Strong indication to proceed with M&E practices that are already in place to record and analyse some outcomes. AMBER: Medium indication to proceed. Recommend discussing the M&E methods with steering group and making a judgement on how to proceed. RED: Not likely to be an effective use of time or resources. Judgement as to how to proceed should be considered with steering group.	A sample of M&E practices from participating children at enrolled ELC settings.	Standardising the analysis of M&E practices already in place at ELC settings will reduce the burden on participating ELC settings in the next stage of evaluation.
5. To what extent is the ***programme acceptable*** to ELC managers and practitioners?	GREEN: analysis of interviews identify little (minor) to no barriers mentioned on supporting the provision of outdoor play and learning within the constructs of the Normalisation Process Theory. AMBER: three or four modifiable barriers identified within the constructs of NPT. RED: several major and non-modifiable barriers identified regarding the acceptability of outdoor play and learning among interviewed participants within the constructs of NPT.	GREEN: Strong indication that outdoor play and learning is becoming a normal practice for ELC practitioners and evaluation of the programme can proceed. AMBER: Indication that outdoor play and learning provision is not yet a normal practice across ELC settings. Findings should be considered with research steering group before proceeding with next stage of evaluation. RED: Doubt regarding whether outdoor play and learning provision is considered a normal practice among ELC practitioners and managers. Steering group should be consulted regarding whether knowledge of outdoor play and learning needs to be promoted among ELC educators before proceeding to an evaluation (e.g., education intervention).	Acceptability of the programme will be assessed through interviews with ELC headteachers/managers and practitioners using the four constructs of NPT (Coherence, Cognitive Participation, Collective Action, Reflexive Monitoring) as part of the process evaluation. See Appendix B for the interview guide.	A focus on major issues associated with the constructs of NPT is considered acceptable for qualitative data rather than quantitative targets.
6. To what extent is the ***study design acceptable*** to ELC managers and practitioners?	GREEN: analysis of interview data identifies little (minor) to no barriers on supporting the study design within the constructs of the NPT. AMBER: three or four modifiable barriers identified within the constructs of NPT. RED: several major and non-modifiable barriers identified regarding the acceptability of the study design among interviewed participants within the constructs of NPT.	GREEN: Strong indication that the study design methods are acceptable among early years educators and children and can be taken forward to a powered effectiveness evaluation. Recommendation as per green target of RQ2. AMBER: Indication that the interview responses should be discussed with the research steering group and a decision made on how the study design can be adapted to be less burdensome. Recommendation as per amber of RQ2. RED: Indication that study design is too labour intensive for participants and needs to be revised before progression to next stage of the evaluation. Recommendation as per red of RQ2.	Acceptability of the study design will be assessed through interviews with ELC headteachers/managers and practitioners using the four constructs of NPT (Coherence, Cognitive Participation, Collective Action, Reflexive Monitoring) as part of the process evaluation. See Appendix B for the interview guide.	A focus on major issues associated with the constructs of NPT is considered acceptable for qualitative data rather than quantitative targets.

**Table 4 ijerph-19-07461-t004:** Outcome measurement methods for the feasibility and pilot study.

Participant	Measurement Tool	Data Collection Timepoints at Participating ELC Settings		
		Baseline (Week 0)	Mid-Point	Follow-Up (Week 5)
Child	Parent/carer Demographic Questionnaire	🗴		
	Strength and Difficulties Questionnaire	🗴		🗴
	Height and weight	🗴		🗴
	Preschool Gross Motor Quality Scale	🗴		🗴
	Axivity (physical activity, sedentary time, sleep)	🗴		🗴
Headteachers/ practitioners	Play behaviours (TOPO)		🗴	🗴
	Semi-structured interviews			🗴
	ELC Monitoring and evaluation tools	🗴		

**Table 5 ijerph-19-07461-t005:** Balance tasks using the preschool gross motor quality scale (adapted from Sun et al., 2010).

Task Code	Balance Task	Criterion Code
B1	Single leg standing	A. Both hands remain on waist. B. Two legs do not lean against each other. C. Non-preferred leg keeps hip extension and knee flexion. D. Preferred leg stands on ground without moving for 5 s.
B2	Tandem standing	A. Both hands remain on waist. B. Postural sway forward and backward less than 30. C. Postural sway side to side less than 30. D. Feet contact ground more than 10 s. E. Feet contact ground more than 20 s.
B3	Walking line forward	A. Walks with each foot contacting the line fully. B. Does not open arms for balance. C. Steps on line precisely without trial. D. Continues heel–toe walking in line for 6 steps. E. Walks with each foot contacting the line fully.
B4	Walking line backward	A. Does not open arms for balance. B. Steps on line precisely without trial. C. Each step goes behind the previous one. D. Walks backward with standard position for 6 steps.

**Table 6 ijerph-19-07461-t006:** Categories of risky play (adapted from [24,25,26]).

Category	Description
*Play with great heights*	Danger of injury from falling from a height relative to the child’s own height such as forms of climbing, jumping, balancing from heights
*Play with high speed*	Uncontrolled speed relative to the child that can lead to a collision with someone (or something) such as on a bicycle, sliding, or running uncontrollably
*Play with dangerous tools*	Tools that can lead to injuries such as knives, hammers, or ropes
*Play near dangerous elements*	Such as a body of water or fire pit
*Rough-and-tumble play*	Where children can harm each other such as wrestling or fencing with sticks
*Play where children go exploring alone*	With the possibility of getting lost such as without supervision and where there are no boundaries or barriers
*Play with impact*	Children crashing into something repeatedly for fun
*Vicarious play*	Children getting excited from watching others engaging in risk

## Data Availability

The quantitative and qualitative data collected will be held in an anonymised format in a UK Research Data Repository. The study’s results will be available to participants, healthcare professionals, the public, and other interested groups via a PhD thesis and Open Access journal articles.

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
