# Peer review of "Evaluating Outdoor Nature-Based Early Learning and Childcare Provision for Children Aged 3 Years: Protocol of a Feasibility and Pilot Quasi-Experimental Design"

_ijerph, 2022, doi:10.3390/ijerph19127461_

Round 1
Reviewer 1 Report
The text presents interesting results of research on the problem of evaluation of play and learning in nature for children in the early learning and care (ELC) sector among Western countries.
The authors made a detailed analysis of the feasibility and acceptability of the program and presented a project to evaluate outdoor games and learning on the example of ELC's activities in Scotland.
In terms of methodology, the text was properly prepared.
The analysis of the literature on the subject is correct.
Congratulations to the authors of the interesting research results.
The only comments I had about the reviewed text are too detailed tables. They could be simplified a bit. However, I decided that the text is very good and the Authors took care of a detailed presentation of the issue, and that the text was recommended for publication without corrections.
However, I think you can include a comment to shorten the data in the tables a bit.
Author Response
Firstly, we would live to thank the reviewer for taking the time to review our manuscript. Our responses have been highlighted.
The text presents interesting results of research on the problem of evaluation of play and learning in nature for children in the early learning and care (ELC) sector among Western countries.
The authors made a detailed analysis of the feasibility and acceptability of the program and presented a project to evaluate outdoor games and learning on the example of ELC's activities in Scotland.
In terms of methodology, the text was properly prepared.
The analysis of the literature on the subject is correct.
Congratulations to the authors of the interesting research results.
The only comments I had about the reviewed text are too detailed tables. They could be simplified a bit. However, I decided that the text is very good and the Authors took care of a detailed presentation of the issue, and that the text was recommended for publication without corrections.
However, I think you can include a comment to shorten the data in the tables a bit.
We thank the reviewer for their feedback. We have amended the tables to reduce their content. Table 1: RQ2 has been removed, RQ3 has been edited. RQ4 has been removed, and RQ 8 has been edited. These changes have also been applied to Table 3.
Figure 1 has also been simplified.
Reviewer 2 Report
This is an exemplary protocol and focused on an important area of public health intervention. I was pleased to see children's learning and behaviour are a focus outcome not simply physical activity and am pleased to see that sleep is measured as a key outcome . In the manuscript the importance of sleep is not discussed- although measured. I would recommend greater attention to the ELC sleep literature and greater focus on sleep measurement (and light exposure) for 3 reasons
- There is considerable evidence that practices in ELC centres affect children' s activity levels, behaviour and learning. I suggest reading the body of work by Dr Sally Staton.and the review article
Light exposure is an important factor in sleep physical activity and behaviour - see
Ulset, V. S., Czajkowski, N. O., Staton, S., Smith, S., Pattinson, C., Allen, A., ... & Bekkhus, M. (2021). Environmental light exposure, rest-activity rhythms, and symptoms of inattention and hyperactivity: An observational study of Australian preschoolers. Journal of Environmental Psychology, 73, 101560. - Sleep evidence rapid change in response to environment. Given this is a feasibility trial of short duration I would recommend much more attention to sleep and light measurement via actigraphy measurement and environment monitoring - within and outside ELC
Author Response
Firstly, we would live to thank the reviewer for taking the time to review our manuscript. Our responses have been highlighted.
- There is considerable evidence that practices in ELC centres affect children' s activity levels, behaviour and learning. I suggest reading the body of work by Dr Sally Staton.and the review article
Light exposure is an important factor in sleep physical activity and behaviour - see
Ulset, V. S., Czajkowski, N. O., Staton, S., Smith, S., Pattinson, C., Allen, A., ... & Bekkhus, M. (2021). Environmental light exposure, rest-activity rhythms, and symptoms of inattention and hyperactivity: An observational study of Australian preschoolers. Journal of Environmental Psychology, 73, 101560.
We thank the reviewer for the recommendation regarding investigating light exposure. Although, light measurement is not a primary component of this study, we are happy to add an exploratory component, of light exposure measured using an Axivity monitor, to our study design. We have added additional information regarding the effect of light exposure on children’s development (line 58 and line 493).
- Sleep evidence rapid change in response to environment. Given this is a feasibility trial of short duration I would recommend much more attention to sleep and light measurement via actigraphy measurement and environment monitoring - within and outside ELC
Thank you for your feedback. We have added an additional sentence acknowledging the association between outdoor play and sleep at Line 45.
Reviewer 3 Report
First of all, I would like to thank the authors for their contribution "Evaluating outdoor nature-based early learning and childcare 2 provision for children aged 3-years: Protocol of a feasibility and 3 pilot randomised and quasi-experimental design".
The paper is well-positioned within the literature and the references are adequate.
The study is well designed and executed.
The used methodology is adequate and appropriate. Please add more info about your protocol SPIRIT.
I propose to make more clear how your results will support policy-making and service delivery in the Scottish ELC sector.
I applaud all the efforts of the author(s) for this research.
Author Response
Firstly, we would live to thank the reviewer for taking the time to review our manuscript. Our responses have been highlighted.
Thank you for your feedback. We have now added a supplementary document (Supplementary Table S1) which contains a table of how this manuscript has followed the SPIRIT 2013 checklist.
Reviewer 4 Report
Thank you for submitting your paper to IJERPH. It was a pleasure reading it. As a practitioner and researcher in innovative developments in environmental education I truly appreciate a review of this kind, as it has the potential to inspire and guide future methodological developments in the field. In my opinion it presents a new perspective to Evaluating outdoor nature-based early learning and childcare provisionthat is interesting and appropriate. But the paper needs to be improved, so I suggest the following major revisions:
The literature review must be orderly: ideas are not connected. I suggest to group ideas and create a logic discourse. The article mixes a wide diversity of experiences in science education. This is confusing and should be carefully defined at the beginning.
Beyond stating that they looked at both formal contexts, what kind of educational approaches and interventions is the research analysing? In other words, what are the criteria to include them in their analysis? Because of the way it is presented, right now it seems it is about ALL kinds of interventions oriented towards design of outdoor nature-based play and learning provision across urban ELC settings in a Scottish metropolitan city, which is really broad and hinders a focused analysis. I suggest thinking carefully about this issue, as this could help being sharper in the analysis and discussion and therefore, to enhance the contributions of the paper.
The purpose has interest, and the manuscript has some merits. However, it has substantial flaws; my main concern is the weakness of the methodology, the lack of information about the instruments and the analysis, or the insufficient information about participants
In terms of research design, In this regard, it seems to me that the views of the experts should be gathered as a medium to help answer a broader research question, which is about the potentials/benefits/tensions of the outdoor nature-based play and learning pro- vision across urban ELC settings in a Scottish metropolitan city, rather than to clarify the views of the experts (because for this, you would need a much bigger sample - this would be just a first exploratory step to identify items.
An elaborated academic discussion of the findings (this section is rather poor now), connecting results to critical issues and trends in the field of evaluating outdoor nature-based early learning and childcare provision (the initial diagnosis). This will probably involve more research work from the authors and more review of existing experiences, in order to critically assess specialists views and contextualise them with other sources of information.
Author Response
Firstly, we would live to thank the reviewer for taking the time to review our manuscript. Our responses have been highlighted.
The literature review must be orderly: ideas are not connected. I suggest to group ideas and create a logic discourse. The article mixes a wide diversity of experiences in science education. This is confusing and should be carefully defined at the beginning.
We thank the reviewer for their feedback. We have attempted to define the flow of the introduction by adding a sentence to Line 35 and throughout the rest of the introduction, clarifying to the reader the role of outdoor nature-based ELC provision on specific outcomes. From this sentence we then present our argument structure as follows:
- Outcome focused reasoning for conducting the research
- ‘Physical’ health outcomes/proxies (PA, sleep, motor competence, balance)
- Emotional and social resilience
- Relationship between affordances of the natural environment and child play behaviours (speaks to differences in exposure across settings).
- The role of the practitioner in facilitating outdoor nature-based play and learning
- Issues associated with the current evidence base
- The Scottish context
- Objectives of the present study
Beyond stating that they looked at both formal contexts, what kind of educational approaches and interventions is the research analysing? In other words, what are the criteria to include them in their analysis? Because of the way it is presented, right now it seems it is about ALL kinds of interventions oriented towards design of outdoor nature-based play and learning provision across urban ELC settings in a Scottish metropolitan city, which is really broad and hinders a focused analysis. I suggest thinking carefully about this issue, as this could help being sharper in the analysis and discussion and therefore, to enhance the contributions of the paper.
We are investigating the feasibility of assessing how nature-based outdoor play and learning is provided across recognised ‘physical’ delivery models of ELC provision within the Scottish ELC sector. This includes fully outdoor models, traditional models, and satellite models. These delivery models are considered, as it pertains to this evaluation, as being operationally different in the types and volume of ‘nature’ the children are exposed to in a daily basis. This includes the affordances of the physical (e.g. forestry, woodland, water access, differing terrain) and social environments (child/practitioner led interactions and opportunities). The principles underpinning our evaluation have been guided by an expert steering group (added info on page 4, line 163) comprising practitioners, managers, and policy makers in the area. Our evaluation, therefore, is being designed to maximise the value and impact of our results for our stakeholders in Scotland. This evaluation will explore the role of the practitioner, child, and interactions therein in the learning and developmental journey of the child, including the underlying educational philosophical positioning of the ELC centre. The intention is determine whether the measurement tools that we are using are sufficient to measure the differences in nature exposure across these different ELC models. The added benefit of this study is that it will provide practitioners and local authority policy makers with preliminary evidence regarding the difference (if any) of the different models of nature exposure across ELC settings on children’s health and wellbeing in this metropolitan city support their provision of funding.
The purpose has interest, and the manuscript has some merits. However, it has substantial flaws; my main concern is the weakness of the methodology, the lack of information about the instruments and the analysis, or the insufficient information about participants
We thank the reviewer for their feedback and will respond to each point individually. For each point the reviewer has highlighted, we have aimed to provide as much information within the manuscript as we considered relevant for the reader to understand why a particular decision was made and how it will be implemented within the study design.
Methodology: From page 3, line 129, we have endeavoured to provide detailed information regarding the choice of our methods and the reasons why, beginning by following the Standard Protocol Items: Recommendations for Interventional Trials (SPIRIT) (see Supplementary Table S1). Table 1 of the manuscript presents each research question, criteria on how the RQ will be addressed, the data collection tools that will address the RQ, the procedures for implementing the tools, details on the study population, and how the data will be analysed to address that particular RQ. This is then reinforced from section 2.8 ‘Measures’ where we provide further information regarding how the data will be collected and address each RQ. Table 3 outlines the progression criteria for each RQ. This allows us the assess the studies results against pre-defined progression criteria, we believe, strengthening our studies methodology. From line 306, page 16, we have provided detailed information regarding how propensity score matching methods will be used in our studies in an effort to create an as-if random study design.
Instruments: From page 23 to 29, line 397 to 555, we have provided detailed information on each outcome measurement tool, and qualitative methods, and how they will be implemented in the present study. Table 1 also demonstrates which instruments will be used to address each RQ. Table 5 illustrates when in the study timeline they will be implemented. With regards to the qualitative methods, we have provided the reader with the interview guide (Appendix A) outlining how the semi-structured interviews will be structured.
Analysis: As demonstrated in section 2.9, page 29 to 30. Analysis of the data will be carried out through three broad stages. Statistical analysis of baseline characteristics and outcome measures. Line 573 to Line 592 describes the descriptive, summary (e.g., median and interquartile ranges), and exploratory statistics (e.g., ANOVA and linear regression) that will be implemented. Qualitative analysis, as detailed from line 593 to 610, will employ thematic analysis methodology and reflexive thinking. Finally, all data collected will be assess against the pre-defined progression criteria as detailed in Table 3. This will support our decision making with regards to what extent each RQ has been addressed and whether progression to a fully powered effectiveness evaluation can take place.
Participants: We have aimed to provide as much information as we deemed sufficient for the reader to understand the study population and the process of identifying and recruiting participants. Section 2.5, from page 10, provides detailed information on the study population and the inclusion and exclusion criteria of participants (Table 3). Section 2.7 provides detailed information about our recruitment strategy and this is support by our study flow diagram, Figure 1.
In terms of research design, In this regard, it seems to me that the views of the experts should be gathered as a medium to help answer a broader research question, which is about the potentials/benefits/tensions of the outdoor nature-based play and learning pro- vision across urban ELC settings in a Scottish metropolitan city, rather than to clarify the views of the experts (because for this, you would need a much bigger sample - this would be just a first exploratory step to identify items.
Thank you for this suggestion. We are using normalisation process theory (see page 15, line 339) within our qualitative interviews with experts such as ELC practitioners, headteachers and managers. This is to address our research question (RQ6) regarding how acceptable the programme is to ELC managers and practitioners. We believe the interview guide that we have developed will allow for themes regarding to benefits and barriers of outdoor nature-based play provision to be discussed.
An elaborated academic discussion of the findings (this section is rather poor now), connecting results to critical issues and trends in the field of evaluating outdoor nature-based early learning and childcare provision (the initial diagnosis). This will probably involve more research work from the authors and more review of existing experiences, in order to critically assess specialists views and contextualise them with other sources of information.
We thank the reviewer for their feedback. We have elaborated on the discussion regarding implications of our findings for the evidence base, policy, and practice as it relates to outdoor nature-based ELC provision.